# Molecular determinants of complexin clamping and activation function

Manindra Bera[1,2], Sathish Ramakrishnan[1,3], Jeff Coleman[1,2], Shyam S Krishnakumar[1,4]*, James E Rothman[1,2]*

[1]Yale Nanobiology Institute, New Haven, United States; [2]Department of Cell Biology, Yale University School of Medicine, New Haven, United States; [3]Department of Pathology, Yale University School of Medicine, New Haven, United States; [4]Departments of Neurology, Yale University School of Medicine, New Haven, United States

**Abstract** Previously we reported that Synaptotagmin-1 and Complexin synergistically clamp the SNARE assembly process to generate and maintain a pool of docked vesicles that fuse rapidly and synchronously upon $Ca^{2+}$ influx (Ramakrishnan et al., 2020). Here, using the same in vitro single-vesicle fusion assay, we determine the molecular details of the Complexin-mediated fusion clamp and its role in $Ca^{2+}$-activation. We find that a delay in fusion kinetics, likely imparted by Synaptotagmin-1, is needed for Complexin to block fusion. Systematic truncation/mutational analyses reveal that continuous alpha-helical accessory-central domains of Complexin are essential for its inhibitory function and specific interaction of the accessory helix with the SNAREpins enhances this functionality. The C-terminal domain promotes clamping by locally elevating Complexin concentration through interactions with the membrane. Independent of their clamping functions, the accessory-central helical domains of Complexin also contribute to rapid $Ca^{2+}$-synchronized vesicle release by increasing the probability of fusion from the clamped state.

**\*For correspondence:**
shyam.krishnakumar@yale.edu (SSK);
james.rothman@yale.edu (JER)

**Competing interest:** The authors declare that no competing interests exist.

## Editor's evaluation

Bera and colleagues revisit several mechanistic questions mainly centered on the accessory helix of mouse complexin (mCpx) and its contribution to the 'fusion clamp' property of mCpx whereby mCpx-SNARE interactions prevent full assembly and subsequent membrane fusion. This clamping function is believed to help generate a metastable pool of release-ready vesicles at the synapse, and it has been studied in a wide variety of systems including mouse, fly, worm, squid, fish, and diverse in vitro biochemical preps over the past ~ 20 years. The authors derive several conclusions from their efforts, but most relevant is a reiteration of a previous proposal that the accessory helix region of mCpx stabilizes a pre-fusion clamped state via interactions with SNAREs.

## Introduction

Neurons communicate with each other at synaptic contacts by releasing neurotransmitters from synaptic vesicles (SVs). This process is tightly controlled by activity-dependent changes in the presynaptic $Ca^{2+}$ concentration and can occur in less than a millisecond after the neuronal spike (*Südhof, 2013*; *Kaeser and Regehr, 2014*). SV fusion is catalyzed by presynaptic SNARE proteins. The SNAREs on the opposing membranes (VAMP2 on the synaptic vesicle membrane; Syntaxin and SNAP25 on the presynaptic plasma membrane) assemble into a four-helix bundle that catalyzes fusion by forcing the two membranes together (*Söllner et al., 1993*; *Weber et al., 1998*). Related SNARE proteins are universally involved in intracellular transport pathways and by themselves can constitutively catalyze

fusion (**Weber et al., 1998**; **McNew et al., 2000**). As such, $Ca^{2+}$-evoked neurotransmitter release occurs from the readily releasable pool (RRP) of vesicles docked/primed at the presynaptic active zone (**Südhof, 2013**; **Kaeser and Regehr, 2014**). The current view is that at a single RRP vesicle, the SNARE complexes are firmly held ('clamped') in a partially assembled state (SNAREpins) close to the point of triggering fusion. Upon $Ca^{2+}$ influx, multiple SNAREpins are synchronously activated to drive ultrafast SV fusion and neurotransmitter release (**Südhof and Rothman, 2009**; **Südhof, 2013**; **Rizo and Xu, 2015**; **Rothman et al., 2017**; **Brunger et al., 2019**).

It is well-established that the late stages of SV fusion are tightly regulated by two synaptic proteins – the presynaptic $Ca^{2+}$ release sensor Synaptotagmin-1 (Syt1) and Complexin (CPX) (**Südhof, 2013**; **Südhof and Rothman, 2009**; **Rizo and Xu, 2015**; **Brunger et al., 2019**). CPX is an evolutionarily conserved cytosolic protein that bind and regulate synaptic SNARE complex assembly (**McMahon et al., 1995**; **Huntwork and Littleton, 2007**; **Martin et al., 2011**; **Trimbuch and Rosenmund, 2016**; **Mohrmann et al., 2015**). Biochemical and biophysical analyses show that CPX promotes the initial stages of SNARE assembly but then blocks complete assembly (**Li et al., 2011**; **Kümmel et al., 2011**; **Lai et al., 2014**; **Krishnakumar et al., 2015**). Thus, it can both facilitate and subsequently inhibit SV fusion. CPX contain distinct domains that mediate the dual clamp/activator function (**Xue et al., 2007**; **Giraudo et al., 2008**; **Trimbuch and Rosenmund, 2016**; **Mohrmann et al., 2015**). The largely unstructured N-terminal domain (residues 1–26 of mammalian CPX1) activates $Ca^{2+}$-regulated vesicular release (**Xue et al., 2010**; **Lai et al., 2016**) while the α-helical accessory domain ($CPX_{acc}$, residues 26–48) serves as the primary clamping domain (**Xue et al., 2007**; **Giraudo et al., 2008**; **Maximov et al., 2009**; **Yang et al., 2010**; **Kümmel et al., 2011**; **Cho et al., 2014**). A central helical sequence within CPX ($CPX_{cen}$, residues 48–70) binds the groove between pre-assembled Syntaxin and VAMP2 and is essential for both function (**Chen et al., 2002**; **Xue et al., 2007**; **Giraudo et al., 2008**; **Maximov et al., 2009**). The remainder c-terminal portion (residues 71–134) has been shown to preferentially associate with curved lipid membrane via an amphipathic helical region and promotes the clamping function (**Kaeser-Woo et al., 2012**; **Wragg et al., 2013**; **Gong et al., 2016**).

The relative strength of CPX facilitatory vs inhibitory activities differs across species (**Yang et al., 2013**; **Trimbuch and Rosenmund, 2016**; **Mohrmann et al., 2015**; **Xue et al., 2009**). As a result of this intricate balance, genetic perturbations of CPX can produce apparently contradictory effects in different systems. For example, knockout (KO) of CPX in neuromuscular synapses of *C. elegans* and *Drosophila* results in increased spontaneous release, decreased evoked release with overall reduction in the RPP size (**Huntwork and Littleton, 2007**; **Cho et al., 2014**; **Martin et al., 2011**; **Hobson et al., 2011**; **Wragg et al., 2013**). In model mammalian synapses, CPX KO abates both spontaneous and evoked release with no significant change in the RRP size (**Reim et al., 2001**; **Xue et al., 2008**; **López-Murcia et al., 2019**) but acute CPX knockdown (KD) reduces synaptic strength, but also increases spontaneous release with a concomitant reduction in the number of primed vesicles (**Maximov et al., 2009**; **Yang et al., 2010**; **Kaeser-Woo et al., 2012**; **Yang et al., 2013**). Some of the apparent discrepancies might be related to the perturbation method used (**Yang et al., 2013**); nonetheless, the physiological role of mammalian CPX in regulating SV fusion and the underlying mechanisms remains in the center of debate (**Mohrmann et al., 2015**; **Trimbuch and Rosenmund, 2016**).

The interpretation of the physiological experiments can be limited by presence of the different CPX isoforms and possible compensatory homeostatic mechanisms. As such, the experiments in live synapses need to be complemented with a reductionist approach where the variables are limited, and the components can be rigorously controlled or altered. It is our hypothesis that the most direct mechanistic insight can be obtained from fully controlled cell-free systems. We have described a biochemically defined fusion setup based on a pore-spanning lipid bilayer setup that is well-suited for this purpose (**Ramakrishnan et al., 2018**; **Ramakrishnan et al., 2019**; **Ramakrishnan et al., 2020**).

Using this in vitro setup, which allows for precision study of the single-vesicle fusion kinetics, we recently demonstrated that mammalian CPX (mCPX), along with Syt1 and SNAREs, are essential and sufficient to achieve $Ca^{2+}$-regulated fusion under physiologically relevant conditions (**Ramakrishnan et al., 2020**). Our data revealed that mCPX and Syt1 act co-operatively to clamp the SNARE assembly process and produce a pool of docked vesicles. The study also revealed that there are at least two types of clamped SNAREpins under a docked vesicle – a small subset that are reversibly clamped by binding to Syt1 (which we termed '*central*') and a larger population that are thought to be free of Syt1 and require mCPX for clamping (termed '*peripheral*'). We further established that Syt1s' ability

to oligomerize and bind SNAREpins via the 'primary' binding site on SNAP25 is key to its ability to clamp *central* SNAREpins and that the activation of these Syt1-associated SNAREpins is sufficient to elicit rapid, $Ca^{2+}$-synchronized vesicle fusion (*Ramakrishnan et al., 2020*).

Building on this work, here we use a systematic in vitro reconstitution strategy to obtain new and direct insights into the molecular basis of mCPX clamping function and its role in establishing $Ca^{2+}$-regulated release. We report that mCPX inhibitory function requires a delay in overall fusion kinetics and involves well-defined interaction of the accessory-central helical fragments with the SNAREpins. The accessory-central helical domains also stimulate $Ca^{2+}$-triggered vesicle fusion from the clamped state. Overall, we find that under physiologically-relevant conditions, mCPX is essential to generate/maintain a pool of docked vesicles and to promote $Ca^{2+}$-triggered rapid ( < 10ms) and synchronous fusion of the docked vesicles.

## Results

To dissect the mCPX clamping functionality, we used physiologically relevant reconstitution conditions similar to our previous work (*Ramakrishnan et al., 2020*). Typically, we used small unilamellar vesicles (SUV) with ~70 copies (outward facing) of VAMP2 (vSUV) without or with ~25 copies Syt1 (Syt1-vSUV) (*Figure 1—figure supplement 1*). We employed pre-formed t-SNAREs (1:1 complex of Syntaxin1 and SNAP-25) in the planar bilayers (containing 15% PS and 3% PIP2) to both simplify the experimental approach and to bypass the requirement of SNARE-assembling chaperones, Munc18 and Munc13 (*Baker and Hughson, 2016*). Mammalian CPX1 (wild type or variants) was included in solution, typically at 2 µM unless noted otherwise (*Figure 1—figure supplement 2*). We used fluorescently labeled lipid (2% ATTO647N-PE) to track docking, clamping and spontaneous fusion of individual vesicles and a content dye (sulforhodamine B) to study $Ca^{2+}$-triggered fusion of docked vesicles from the clamped state.

To focus on the 'clamping' of constitutive fusion events, we monitored large ensembles of vesicles to determine the percent remaining unfused as a function of time elapsed after docking and quantified as 'survival percentages' (*Ramakrishnan et al., 2019*; *Ramakrishnan et al., 2020*; *Ramakrishnan et al., 2018*). Docked immobile vesicles that remained un-fused during the initial 10-min observation period were defined as 'clamped' and the 'docking-to-fusion' delay enabled us to quantify the strength of the fusion clamp (*Ramakrishnan et al., 2019*; *Ramakrishnan et al., 2020*; *Ramakrishnan et al., 2018*). Since we track the fate of single vesicles, this analysis allowed us to examine the 'clamping' mechanism, independent of any alteration in the preceding docking sub-step.

Our earlier results showed that Syt1 alone can meaningfully delay but not stably clamp SNARE-mediated fusion. Similarly, mCPX, on its own, is ineffective in clamping SNARE-driven vesicle fusion. In fact, both Syt1 and mCPX are needed to produce a stably 'clamped' state which can then be reversed by $Ca^{2+}$ (*Ramakrishnan et al., 2020*). It is possible that Syt1 and mCPX1 either act jointly to generate a new intermediate state in the SNARE assembly pathway or operate sequentially, with the kinetic delay introduced by Syt1 enabling mCPX to arrest SNARE assembly. To distinguish between these possibilities, we developed a mimic for the Syt1 clamp – a lipid-conjugated ssDNA that is capable of regulating SNARE-driven fusion in situ. Without directly interacting with the SNAREs, the specific base-pair hybridization of the complementary ssDNA reconstituted into the SUVs and the planar bilayer introduces a steric barrier which is expected to, and indeed does delay fusion (*Figure 1*, *Figure 1—figure supplement 3*). Moreover, this docking-to-fusion delay could be varied by adjusting the number of ssDNA molecules (*Figure 1—figure supplement 3*).

We then assessed the effect of mCPX on ssDNA-regulated fusion of vSUV in the absence of $Ca^{2+}$ (*Figure 1*). mCPX was able to near-completely arrest spontaneous fusion of vSUV to generate stably docked vesicles, provided that the rate of SNARE-mediated fusion was sterically delayed by ~20 copies of ssDNA (*Figure 1*, *Figure 1—figure supplement 4*). The majority of the vSUVs were immobile following docking to the t-SNARE-containing suspended bilayer (*Figure 1*, *Figure 1—figure supplement 4*), and they rarely fused over the initial observation period. In contrast, little or no inhibition was observed in control experiments with ~5 copies of ssDNA that did not introduce a detectable delay in the fusion process, as all docked vesicles proceeded to fuse spontaneously typically within 1–2 s (*Figure 1*, *Figure 1—figure supplement 4*). This suggests that it is the delay in fusion per se that is necessary for the mCPX inhibitory function, and importantly that the mCPX clamp is not dependent or influenced by the ssDNA molecules (*Figure 1—figure supplement 4*). Thus, our data indicates

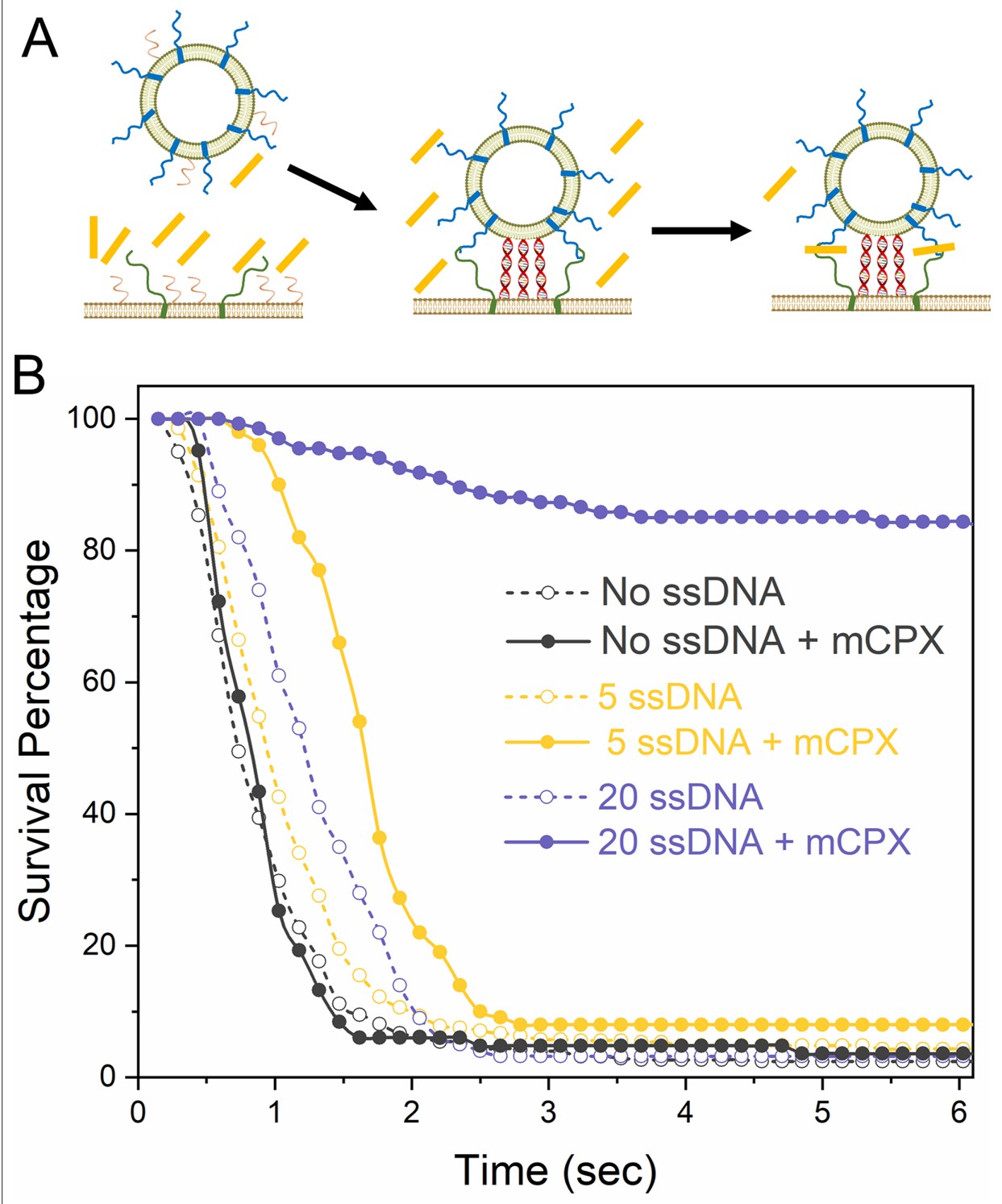

**Figure 1.** Syt1 and mCPX act sequentially to arrest SNARE-driven fusion. (**A**) Schematic of the programmable DNA-based mimetic used to simulate the Syt1 clamp on the SNARE-driven fusion. Annealing of the complementary ssDNA reconstituted into the SUV and the bilayer in dsDNA sterically counters the polarized SNARE assembly process and introduces a docking-to-fusion delay reminiscent of Syt1 (**B**) Survival analysis (Kaplan-Meier plot) curve shows that a nominal dock-to-fusion delay introduced by 20 copies of ssDNA (purple) allows mCPX to arrest spontaneous fusion of vSUVs. In contrast, no clamping was observed with 5 copies of ssDNA (yellow) which created no appreciable delay in the fusion kinetics. This suggests a sequentially mode of action for Syt1 and mCPX, wherein the kinetic delay introduced by Syt1 enables mCPX to block SNARE-driven fusion. Data was

*Figure 1 continued on next page*

*Figure 1 continued*

obtained from a minimum of three independent experiments, with at least 100 vesicles analyzed for each condition. A representative survival curve is shown for clarity.

The online version of this article includes the following source data and figure supplement(s) for figure 1:

**Source data 1.** Data and summary statistics for DNA-regulated fusion assay.

**Figure supplement 1.** Coomassie-stained SDS-PAGE analysis of the proteins used in this study.

**Figure supplement 2.** End-point (10 min) survival analysis shows that the 5 min pre-incubation of mCPX (2 µM) with either the t-SNARE containing bilayer (orange bar) or Syt1-vSUVs (green bar) does not affect its overall clamping ability.

**Figure supplement 2—source data 1.** Data and summary statistics of CPX incubation analysis.

**Figure supplement 3.** DNA-hybridization regulates SNARE-mediated membrane fusion.

**Figure supplement 3—source data 1.** Data for DNA regulation bulk fusion assay.

**Figure supplement 4.** Delay in fusion kinetics required for CPX clamping.

**Figure supplement 4—source data 1.** Data and summary statistics of optimizing the DNA-regulated fusion assay.

that Syt1 and mCPX likely act sequentially to produce a synergistic clamp, with the delay introduced by Syt1 meta-stable clamp enabling CPX to bind and block the full assembly of the SNARE complex.

Next, we investigated the role of the distinct domains of mCPX in establishing the fusion block using Syt1 containing vSUV (Syt1-vSUVs). On their own, a majority (~80%) Syt1-vSUVs that docked to the t-SNARE containing bilayer surface were mobile and fused on an average 5–6 s after docking, while a small fraction (~20%) were immobile and stably clamped (*Figure 2A and B*). Inclusion of 2 µM wild-type mCPX (mCPX$^{WT}$) enhanced the vesicle docking rate, with an ~ three-fold increase in the total number of stably docked vesicles and >95% of Syt1-vSUVs remaining immobile post-docking (*Figure 2A and B*). This is consistent with our earlier findings (*Ramakrishnan et al., 2020*). A truncation mutant (mCPX$^{26-134}$) lacking the unstructured N-terminal domain had very little or no effect on the vesicle docking rate or the fusion clamp, with vesicle behavior near identical to CPX$^{WT}$ (*Figure 2A and B*). Deletion of the CPX$_{acc}$ in addition to N-terminal domain (mCPX$^{48-134}$) increased the number of docked vesicles (~ two-fold) but abrogated the inhibitory function with majority of the docked vesicles proceeding to fuse spontaneously (*Figure 2A and B*). Targeted mutations in CPX$_{cen}$ (R48A Y52A K69A Y70A; mCPX$^{4A}$) that disrupt its interaction with the SNAREpins completely abolished both the stimulatory effect on vesicle docking and the fusion clamp (*Figure 2A and B*). In fact, both the CPX$_{acc}$ deletion (mCPX$^{48-134}$) and CPX$_{cen}$ modifications (mCPX$^{4A}$) resulted in complete loss of mCPX inhibitory function and could not be rescued even at highest concentration (20 µM) tested (*Figure 2C*, *Figure 2—figure supplement 1*). Deletion of the c-terminal domain (mCPX$^{26-83}$) lowered the clamping efficiency (*Figure 2A and B*) with ~50% vesicles clamped under the standard experimental conditions (2 µM mCPX$^{26-83}$). However, the inhibitory function was rescued simply by raising the concentration and was completely restored at 20 µM mCPX$^{26-83}$ (*Figure 2C*, *Figure 2—figure supplement 1*).

Altogether, we conclude that the CPX$_{cen}$-SNAREpin interaction promotes vesicle docking, and this interaction along with CPX$_{acc}$ are critical for mCPX mediated clamping under physiologically relevant experimental conditions. The c-terminal domain plays an auxiliary role and contributes to the mCPX inhibitory function likely by concentrating it on vesicle surfaces due to its curvature-binding region. Supporting this, a CPX mutant (CPX$^{L117W}$) that enhances the curved membrane association of the c-terminal domain (*Seiler et al., 2009*) increased the clamping efficiency as compared to CPX$^{WT}$ (*Figure 2—figure supplement 2*).

Biophysical and structural studies have demonstrated that binding of the CPX$_{cen}$ to the SNAREpins positions the CPX$_{acc}$ to effectively block complete SNARE assembly (*Kümmel et al., 2011*; *Giraudo et al., 2008*; *Krishnakumar et al., 2015*). While the precise mode of action is under debate, there is evidence that this involves specific interactions of CPX$_{acc}$ with the c-terminal region of the SNAREpins (*Kümmel et al., 2011*; *Malsam et al., 2020*). Critical information about these inter-molecular interactions was provided by the X-ray structure of mCPX bound to a mimetic of a pre-fusion half-zippered SNAREpins (*Kümmel et al., 2011*). It revealed that the CPX$_{cen}$ is anchored to one SNARE complex, while its CPX$_{acc}$ extends away and binds to the t-SNARE in a second SNARE complex in a site normally occupied by the C-terminus of the VAMP2 helix (*Kümmel et al., 2011*; *Krishnakumar et al., 2015*).

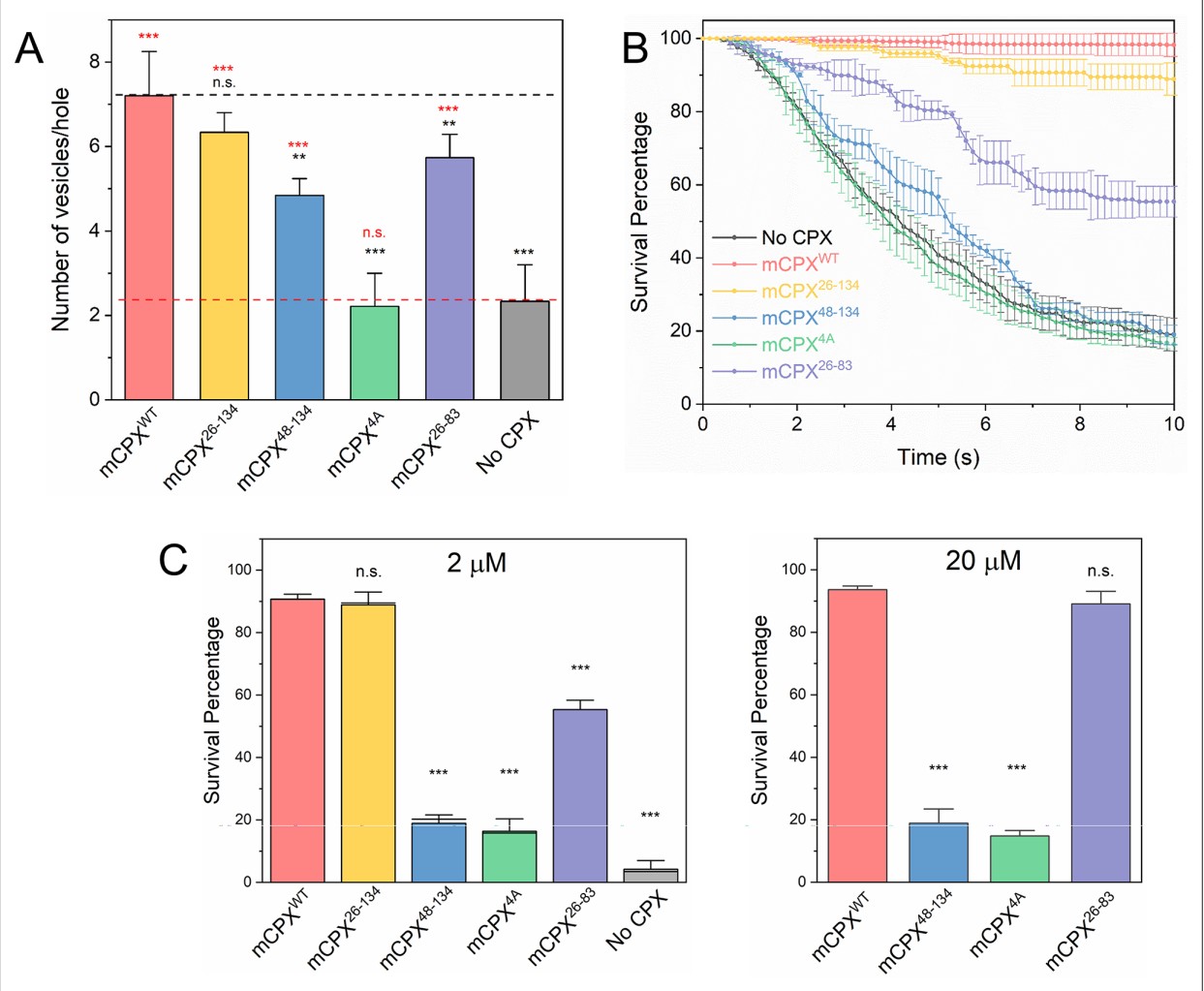

**Figure 2.** Molecular determinants of Complexin clamping function. The effect of mCPX mutants on docking and clamping of spontaneous fusion was assessed using a single-vesicle analysis with a pore-spanning bilayer setup. (**A**) Inclusion of mCPX increases the number of docked Syt1-vSUVs and this stimulatory effect is greatly reduced when the interaction of the CPX$^{cen}$ to the SNAREpins is disrupted targeted mutations (mCPX$^{4A}$). In contrast, deletion of the N-terminal domain (CPX$^{26-134}$) or accessory helix (CPX$^{48-134}$) or the c-terminal portion (CPX$^{26-83}$) exhibit limited effect of the vesicle docking. In all cases, a mutant form of VAMP2 (VAMP2$^{4X}$) which eliminated fusion was used to unambiguously estimate the number of docked vesicles after the 10 min interaction phase. (**B**) The time between docking and fusion was measured for each docked vesicle and the results for the whole population are presented as a survival curve (Kaplan-Meier plots). Syt1-vSUVs (black curve) are diffusively mobile upon docking and fuse spontaneous with a half-time of ~5 s. Addition of soluble mCPX (red curve) fully arrest fusion to produce stably docked SUVs that attach and remain in place during the entire period of observation. CPX mutants with impaired SNARE interaction (mCPX$^{4A}$, green curve) or lacking the accessory helical domain (mCPX$^{48-134}$, yellow curve) fail to clamp fusion whilst the removal of c-terminal portion (mCPX$^{26-83}$, purple curve) produces a partial clamping phenotype. The N-terminal domain is not involved in establishing the fusion clamp (**C**) End-point analysis at 10 s post-docking shows that the both the accessory helix deletion (mCPX$^{48-134}$) and CPX$_{cen}$ modifications (mCPX$^{4A}$) result in complete loss of inhibitory function and cannot be rescued even at 20 μM concentration. In contrast, the clamping function of the c-terminal deletion mutant (mCPX$^{26-83}$) is fully restored at high CPX concentration. The average values and standard deviations from three independent experiments (with ~300 vesicles in total) are shown. **p < 0.01; *** p < 0.001 using the Student's t-test.

The online version of this article includes the following source data and figure supplement(s) for figure 2:

**Source data 1.** Data and summary statistics of docking and survival analysis for CPX mutants.

**Figure supplement 1.** Survival analysis (Kaplan-Meier plots) of Syt1-vSUVs shows that the loss of clamping phenotypes observed with CPX mutant with impaired SNARE interaction (mCPX$^{4A}$, green curve) or lacking the accessory helical domain (mCPX$^{48-134}$, blue curve) is not rescued at high (20 μM) CPX concentrations.

**Figure supplement 1—source data 1.** Data and statistics of survival analysis of CPX mutants at high concentration.

**Figure supplement 2.** Dose-dependency analysis using Syt1-vSUVs shows that CPX mutant with a hydrophobic mutation (mCPX$^{L117W}$, red curve) designed to improve its membrane association is more efficient in clamping fusion as compared to the CPX$^{WT}$ (black curve).

**Figure supplement 2—source data 1.** Data for titration analysis for CPX L117W mutant.

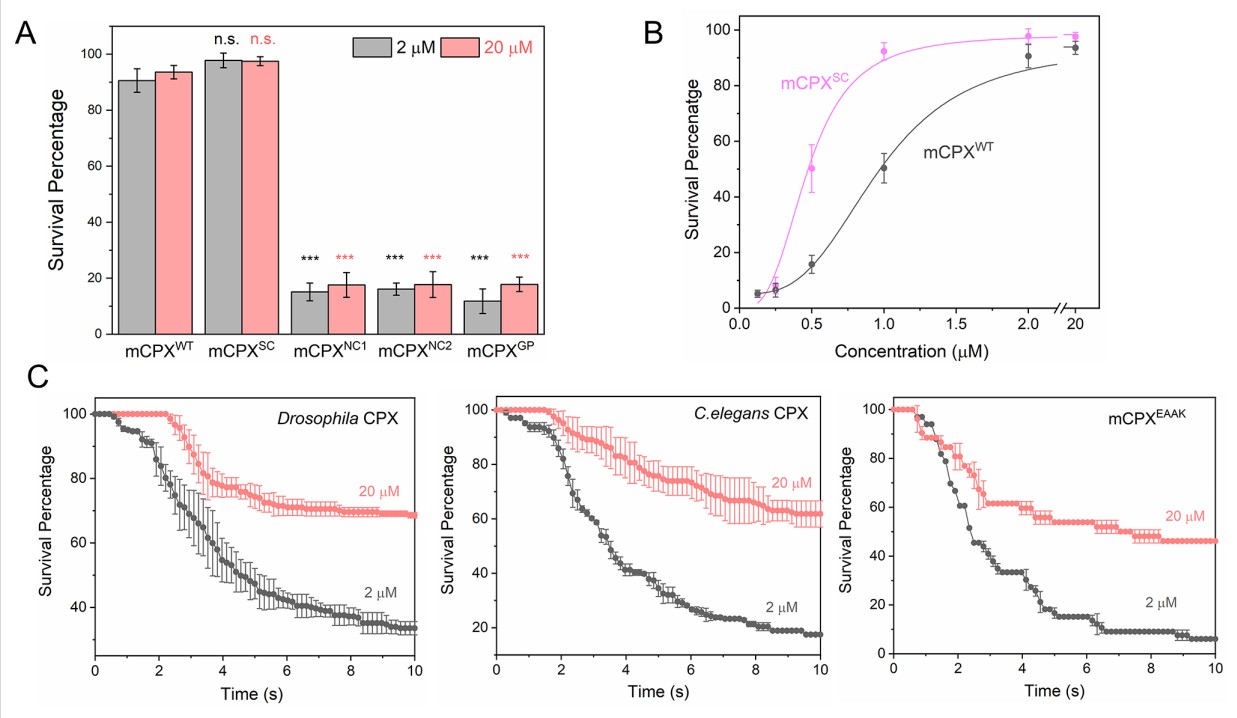

**Figure 3.** Specific interaction of mCPX accessory helix with SNAREs enhances its clamping function. (**A**) End-point survival analysis (measured at 10 s post docking) using Syt1-vSUVs demonstrates that disrupting the binding of the CPX$_{acc}$ to either the t-SNAREs (CPX$^{NC1}$) or the VAMP2 (CPX$^{NC2}$) abrogates the clamping function, and that a helix breaking mutation (CPX$^{GP}$) introduced between CPX$_{cen}$ and CPX$_{acc}$ also abrogates the fusion clamp. (**B**) In contrast, mutations designed to enhance the binding of CPX$^{acc}$ to t-SNAREs (CPX$^{SC}$) increase the potency of the CPX clamp. This indicates efficient clamping by CPX requires a continuous rigid helix along with specific interaction of the CPX$_{acc}$ with the assembling SNARE complex. (**C**) Supporting this notion, survival analysis (Kaplan-Meier plots) shows that both *Drosophila* and *C. elegans* CPXs, which have very low sequence identity with the mCPX accessory domain, and a CPX mutant with a randomized accessory helical sequence (CPX$^{EAAK}$) have poor clamping efficiency under standard (2 μM) experimental conditions and only partial clamping at higher (20 μM) concentration. The average values and standard deviations from three to four independent experiments (with ~250 vesicles in total) are shown. *** indicates p < 0.001 using the Student's t-test.

The online version of this article includes the following source data and figure supplement(s) for figure 3:

**Source data 1.** Data and summary statistics of survival analysis for ceCPX, dmCPX and mCPX mutants.

**Figure supplement 1.** Survival analysis (Kaplan-Meier plots) using Syt1-vSUVs and 2 μM CPX shows targeted mutations that disrupt the interaction of CPX$_{acc}$ with the t-SNARE (mCPX$^{NC1}$, blue curve) or VAMP2 (mCPX$^{NC2}$, green curve) abrogate the clamping function.

**Figure supplement 1—source data 1.** Data and summary statistics of survival analysis of mCPX mutants.

**Figure supplement 2.** Docking analysis with Syt1-vSUVs show that CPX$_{acc}$ does not contribute significantly to stimulatory effect on vesicle docking.

**Figure supplement 2—source data 1.** Data and summary statistics of docking analysis for ceCPX, dmCPX and mCPX mutants.

**Figure supplement 3.** Sequence alignment of alpha helical CPX$_{acc}$-CPX$_{cen}$ portion (mCPX residues 26–70) shows that the CPX$_{cen}$ is largely conserved while CPX$_{acc}$ is highly divergent across different species.

**Figure supplement 4.** Syt1-vSUVs stably clamped at high concentration (20 μM) of *Drosophila* CPX (dmCPX), C.

**Figure supplement 4—source data 1.** Data for calcium sensitivity of ceCPX, dmCPX and mCPX mutant.

This *trans*-insertion model suggest a straightforward mechanism by which CPX$_{acc}$ can block the complete assembly of the SNARE complex (*Kümmel et al., 2011*; *Krishnakumar et al., 2015*).

To ascertain if the hydrophobic CPX$_{acc}$-t-SNARE binding interfaces observed in the crystal structure are involved in clamping in our in vitro system, we tested known CPX mutants designed to either enhance (D27L E34F R37A, 'super-clamp' mutant mCPX$^{SC}$) *or* weaken (A30E A31E L41 A44E, 'non-clamp' mutant 1 mCPX$^{NC1}$) this interaction (*Giraudo et al., 2009*; *Kümmel et al., 2011*). Survival analysis of Syt1-vSUVs showed that the binding interface mutants indeed alter the inhibitory activity of CPX as predicted (*Figure 3A*, *Figure 3—figure supplement 1*). The mCPX$^{NC1}$ abrogated the fusion clamp and was inactive even at higher (20 μM) concentration (*Figure 3A*, *Figure 3—figure supplement 1*). In contrast, mCPX$^{SC}$ increased the clamping efficiency and produced stably docked vesicles

at lower concentrations (IC$_{50}$ ~0.5 μM) compared to the mCPX$^{WT}$ (IC$_{50}$ ~1 μM) (*Figure 3B*, *Figure 3—figure supplement 1*). These findings strongly support the notion that the CPX$_{acc}$-t-SNARE interactions observed in the pre-fusion mCPX-SNAREpin crystal is relevant for the CPX clamping function and is physiologically relevant.

Another key feature of the pre-fusion crystal structure is that the mCPX helix (CPX$_{cen}$ +CPX$_{acc}$) forms a rigid bridge between two SNARE complexes (*Kümmel et al., 2011*; *Krishnakumar et al., 2015*). To test whether the rigidity of mCPX is important for clamping, we used a mCPX mutant (mCPX$^{GP}$) having a helix-breaking linker (GPGP) inserted between CPX$_{cen}$ and CPX$_{acc}$. We found that disrupting the continuous helix indeed reduced the clamping efficiency (*Figure 3A*, *Figure 3—figure supplement 1*) indicating that the continuity and rigidity of the CPX helix is mechanistically important for its inhibitory function. This is also consistent with other previous studies (*Chen et al., 2002*; *Xue et al., 2007*; *Cho et al., 2014*; *Radoff et al., 2014*).

Recently, site-specific photo-crosslinking studies in a reconstituted fusion assay revealed that CPX$_{acc}$ (of closely related mammalian isoform CPXII) binds to the c-terminal portions of SNAP25 and VAMP2 and both interactions are important for the mCPX inhibitory function (*Malsam et al., 2020*). The binding interface for SNAP25 was nearly identical to CPX$_{acc}$-t-SNARE interface observed in the crystal structure while the opposite side of the CPX$_{acc}$ was found to interact with VAMP2 (*Malsam et al., 2020*). Note that this portion of VAMP2 was missing in the pre-fusion SNAREpin mimetic used for in the crystal structural analysis (*Kümmel et al., 2011*). To understand if the aforementioned CPX$_{acc}$-VAMP2 interaction is also part of the clamping mechanism in our cell-free system, we used a mCPX mutant (K33E R37E A40K A44E; non-clamp mutant 2, mCPX$^{NC2}$) that reverses the charge on key binding residues and is thus expected to disrupt this interaction (*Malsam et al., 2020*). mCPX$^{NC2}$ also failed to clamp spontaneous fusion of Syt1-vSUVs in our in vitro assay (*Figure 3A*, *Figure 3—figure supplement 1*) and was phenotypically analogous to the t-SNARE non-binding mutant (mCPX$^{NC1}$). This indicated the CPX$_{acc}$ interacts with both t- and v-SNAREs to block full-zippering. As expected, because their central helix is unaltered, the majority of CPX$_{acc}$ mutants tested retained the ability to promote vesicle docking process albeit lower than mCPX$^{WT}$ (*Figure 3—figure supplement 2*).

CPX$_{cen}$ is broadly conserved with ~75% amino acid sequence identity across diverse species, whereas CPX$_{acc}$ is highly divergent with ~25% sequence identity (*Figure 3—figure supplement 3*). Nonetheless, cross-species rescue experiments have been largely successful, and in fact, CPX$_{acc}$ could be exchanged without impairing function in mammalian, fly and nematode synapses (*Xue et al., 2009*; *Cho et al., 2014*; *Radoff et al., 2014*). This raises the question whether the distinct CPX$_{acc}$-SNARE interactions that are vital for mCPX inhibitory functionality in our in vitro assays are physiologically relevant. To address this, we examined the clamping ability of the *C. elegans* (ceCPX) and *Drosophila* (dmCPX) orthologs of mCPX in our in vitro reconstituted assay. Under standard experimental conditions (2 μM CPX), both ceCPX and dmCPX were able to promote vesicle docking (*Figure 3—figure supplement 2*) but were considerably less efficient (~15% and ~ 30%, respectively) in preventing spontaneous fusion of Syt1-vSUV (*Figure 3C*) as compared near-complete ( > 95%) fusion clamp observed with mCPX (*Figure 2B and C*). Interestingly, simply increasing the concentrations improved the clamping efficacy of both dmCPX and ceCPX, with ~60–70% of docked vesicles stably-clamped at 20 μM concentration (*Figure 3C*) and remained Ca$^{2+}$-sensitive (*Figure 3—figure supplement 4*).

This suggests that specific molecular interactions of CPX$_{acc}$ with SNAREs likely increase the potency of the mCPX inhibitory function and that this effect may be occluded at high concentrations of CPX. To verify this, we examined the effect of the mCPX mutant wherein the endogenous CPX$_{acc}$ domain (residues 26–48) is replaced with an artificial alpha helix based on a Glu-Ala-Ala-Lys (EAAK) motif repeated seven times (*Radoff et al., 2014*). Noteworthy, this construct (mCPX$^{EAAK}$) was able to fully-restore CPX inhibitory functionality in *C. elegans* neuromuscular synapses (*Radoff et al., 2014*). In our in vitro assay, CPX$^{EAAK}$ enhanced initial docking (*Figure 3—figure supplement 2*) but failed to clamp spontaneous fusion (~10% efficiency) under standard experimental conditions (2 μM CPX) and was moderately effective (~50% efficiency) at higher (20 μM CPX) concentration (*Figure 3C*, *Figure 3—figure supplement 4*). We note that the accessory helix of mCPX$^{EAAK}$ is more hydrophobic in nature and interestingly resembles couple of the gain-of-function 'super-clamp' mutations with residue Asp-27 and Glu-34 replaced with Ala (*Figure 3—figure supplement 3*). This could potentially explain mCPX$^{EAAK}$ ability to partially clamp vesicle fusion at high (20 μM) CPX concentration. Overall, our data

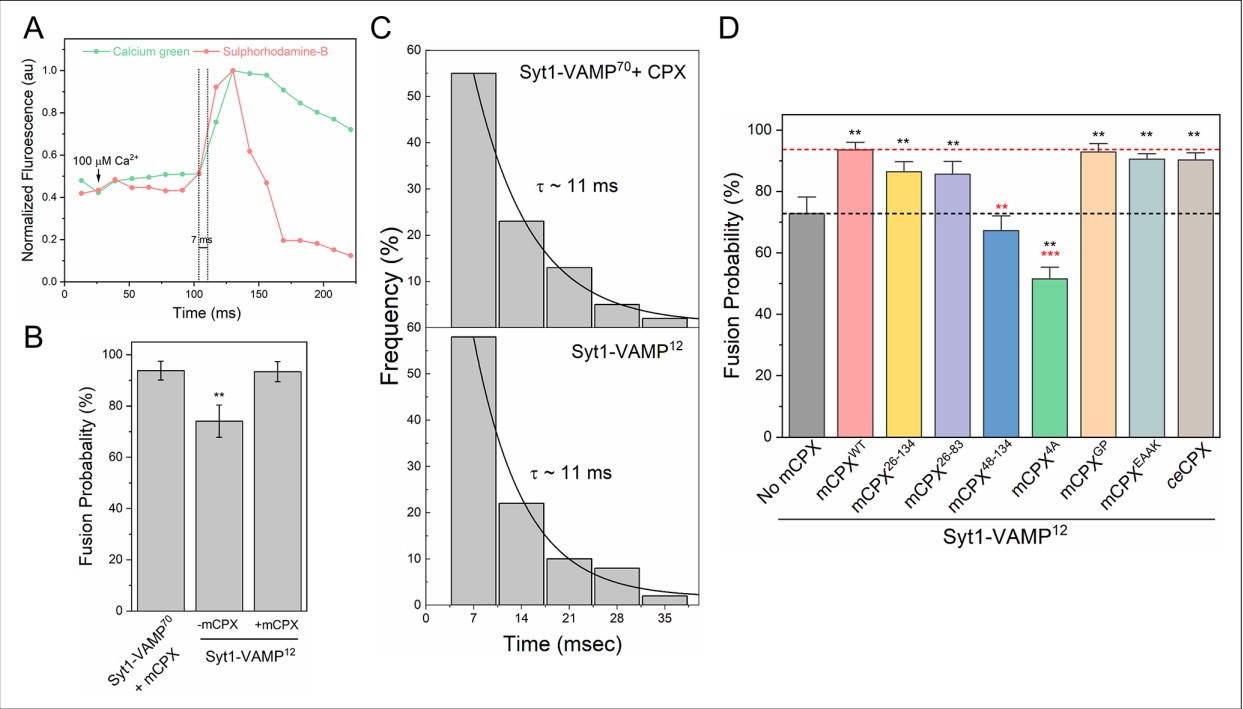

**Figure 4.** Complexin increases the probability of $Ca^{2+}$-triggered vesicular release. (**A**) The effect of mCPX on $Ca^{2+}$-triggered fusion was assessed using a content-release assay with Sulforhodamine-B loaded vesicles. Sulforhodamine-B is largely self-quenched when encapsulated inside an SUV. Fusion of the vesicle results in dilution of the probe, which is accompanied by increasing fluorescence. The $Ca^{2+}$-sensor dye, Calcium Green, introduced in the suspended bilayer (via a lipophilic 24-carbon alkyl chain) was used to monitor the arrival of $Ca^{2+}$ at/near the docked vesicles. A representative fluorescence trace before and after the addition of 100 µM $Ca^{2+}$ shows that the rise in Sulforhodamine-B (red curve) fluorescence intensity occurs within a single frame (13 ms) of $Ca^{2+}$ binding to local Calcium green (green curve) (**B**) End-point analysis at 1 min post $Ca^{2+}$-addition shows that >90% of all Syt1/mCPX-clamped vesicles (~70 copies of VAMP2 and ~25 copies of Syt1) fuse following $Ca^{2+}$ addition as compared to ~70% of Syt1-clamped vesicles (~13 copies of VAMP2 and ~25 copies of Syt1). Inclusion of mCPX enhances the fusion probability even under the low-VAMP2 condition suggesting that mCPX promote $Ca^{2+}$-triggered fusion independent of its clamping function. (**C**) Kinetic analysis shows that the clamped vesicles with or without mCPX fuse rapidly following $Ca^{2+}$-addition with near identical time constant of ~11 ms. This represents the temporal resolution limit of our recordings (13 ms frame rate) and the true $Ca^{2+}$-triggered fusion rate may well be below 10 ms. (**D**) Deletion and mutational analysis under low-VAMP2 conditions (SUVs with ~13 copies of VAMP2 and ~25 copies of Syt1) show that the deletion of CPX_acc (CPX48-134, blue bar) or disruption of CPX_cen-SNARE interaction (CPX4A, green bar) abrogate the stimulatory function, but deletion of the N-terminal portion (CPX26-134, yellow bar) or the c-terminal domain (CPX26-83, purple bar) has no effect. The stimulatory function does not require rigid CPX_acc-CPX_cen helix (mCPXGP, orange bar) nor clamping specific CPX_acc-SNARE interaction as non-clamping CPXEAAK mutant (cyan bar) and *C. elegans* ortholog (ceCPX, brown bar) retain stimulatory function. The average values and standard deviations from three independent experiments (with ~100 vesicles in total) are shown. ** p < 0.01, ***p < 0.001 using the Student's t-test.

The online version of this article includes the following source data for figure 4:

**Source data 1.** Data and summary statistics of effect of mCPX mutants on calcium activation of fusion.

---

supports the notion the specific CPX_acc-SNARE interaction is functionally relevant and likely enhances CPX inhibitory function.

Finally, we evaluated the probability and rate of $Ca^{2+}$-triggered fusion from the clamped state in the presence and absence of mCPX. We used Syt1-vSUV loaded with Sulforhodamine B (fluorescent content marker) to track full-fusion events and lipid-conjugated $Ca^{2+}$ indicator (Calcium green C24) attached to the planar bilayer to estimate the time of arrival of $Ca^{2+}$ at/near the docked vesicles (*Figure 4A*). Consistent with our previous study, the influx of free $Ca^{2+}$ (100 µM) triggered simultaneous fusion of >90% of the Syt1/mCPX-clamped vesicles (*Figure 4B*). These vesicles fused rapidly and synchronously, with a characteristic time-constant ($\tau$) of ~11 msec following the arrival of $Ca^{2+}$ locally (*Figure 4C*). Considering that the majority of $Ca^{2+}$-triggered fusion occurs within a single frame (13 ms), we suspect that the true $Ca^{2+}$-driven fusion rate is likely <10 ms.

In absence of mCPX, we observed a relatively small number of docked vesicles prior to $Ca^{2+}$ influx and this precluded any meaningful quantitative analysis. Hence, to obtain stably docked vesicles without mCPX, we used low VAMP2 conditions that is, SUVs containing ~13 copies of VAMP2 and ~25

copies of Syt1 (*Figure 1—figure supplement 1*). We have previously demonstrated that under these conditions, Syt1 alone is sufficient to produce stably-clamped vesicles (*Ramakrishnan et al., 2019*) and that is what we observe, with >95% of docked vesicles immobile post-docking. Addition of $Ca^{2+}$ (100 µM) triggered rapid and synchronous fusion of ~70% of these Syt1-clamped vesicles (with $\tau$ ~ 11 msec) as compared to >90% fusion of Syt1/mCPX-clamped vesicles (*Figure 4B and C*). Besides mCPX, the number of SNAREpins available on a given vesicle is also different between the two conditions (~13 VAMP2 in Syt1-alone vs. ~70 VAMP2 in Syt1/CPX). Hence, to verify that the observed effect is directly attributable to mCPX, we tested and confirmed that the inclusion of mCPX under low VAMP2 conditions increased the $Ca^{2+}$-triggered fusion probability (~90%) from the clamped state (*Figure 4B*). This indicated that besides clamping vesicle fusion, mCPX also promotes $Ca^{2+}$-triggered vesicle fusion from the clamped state.

To identify the molecular aspects underlying mCPX stimulatory function, we examined the effect of mCPX mutants on $Ca^{2+}$-triggered release under low VAMP2 conditions (*Figure 4D*). Deletion of the N-terminal alone (mCPX$^{26-134}$) or the N- and C-terminal domains (mCPX$^{26-83}$) had very little or no effect on the mCPX stimulatory function (*Figure 4D*). However, deletion of the $CPX_{acc}$ in addition to N-terminal domain (mCPX$^{48-134}$) or disrupting the $CPX_{cen}$-SNARE interaction (mCPX$^{4A}$) abrogated the mCPX activation function (*Figure 4D*) suggesting that the $CPX_{cen}$ and $CPX_{acc}$ domains are crucial for mCPX's stimulatory function. In contrast to their clamping function, disrupting the rigidity and continuity of the $CPX_{cen}$-$CPX_{acc}$ helix with the GPGP insert (mCPX$^{GP}$) had no effect on the activation function (*Figure 4D*). Furthermore, the mCPX mutant with a randomized $CPX_{acc}$ (CPX$^{EAAK}$) and the *C. elegans* ortholog (ceCPX), both of which lack the clamping functionality under the experimental conditions (*Figure 3*), retained the ability to promote $Ca^{2+}$-triggered fusion of the docked vesicles (*Figure 4D*). Taken together, our data suggest that specific interactions of $CPX_{cen}$ with SNAREpins are required for the mCPX stimulatory function and the $CPX_{acc}$ can act independently of $CPX_{cen}$ via a mechanism different from that involved in clamping vesicle fusion.

## Discussion

Our data indicates the mCPX is critical to produce the 'clamped' state and also contribute towards synchronizing fusion to $Ca^{2+}$ influx. In addition, we find that the stimulatory and clamping functionality of mCPX are mechanistically separable. There is a long-standing debate over the role of CPX in establishing a fusion clamp and perhaps the best evidence in support has come from biochemical analyses (*Giraudo et al., 2006*; *Giraudo et al., 2008*; *Kümmel et al., 2011*; *Lai et al., 2014*) and physiological studies in invertebrate synapses (*Huntwork and Littleton, 2007*; *Cho et al., 2014*; *Martin et al., 2011*; *Hobson et al., 2011*). In the case of mammalian synapses, a role for CPX in blocking spontaneous release events remains controversial because KD/KO manipulations yield seemingly contradictory results and show neuron-specific differences (*Xue et al., 2008*; *López-Murcia et al., 2019*; *Maximov et al., 2009*; *Yang et al., 2013*). Here, using a fully defined albeit simplified cell-free system we provide compelling evidence that mCPX is an integral part of the overall clamping mechanism and delineate the molecular mechanism of mCPX inhibitory function. The distinct effects of different CPX truncation and targeted mutations match with data obtained from other reductionist or even physiological systems (*Giraudo et al., 2006*; *Giraudo et al., 2008*; *Kümmel et al., 2011*; *Cho et al., 2014*; *Lai et al., 2014*; *Gong et al., 2016*) forcefully arguing for the physiological relevance of results obtained from our in vitro reconstituted assay.

Our experiments indicate that mCPX inhibitory function entails distinct and specific interactions of the $CPX_{cen}$ and $CPX_{acc}$ domains with assembling SNAREpins, and that the c-terminal domain augments clamping function by increasing the local concentration and/or by proper orientation of CPX via interactions with the vesicle membrane (*Figure 2*). Our results indicate that $CPX_{cen}$ binds in the groove between assembling Syntaxin and VAMP2 helices at the early stages of vesicle docking to stabilize the partially-zippered SNAREpins, consequently promote vesicle docking. This in turn positions $CPX_{acc}$ to block further zippering of SNARE complex both by directly capturing the VAMP2 c-terminus and by simultaneously occupying its binding pocket on the t-SNARE. In line with earlier reports (*Chen et al., 2002*; *Xue et al., 2007*; *Radoff et al., 2014*; *Cho et al., 2014*; *Kümmel et al., 2011*), we find that a continuous, rigid CPX helix is essential for a stable fusion clamp. However, the precise configuration of this clamped state under the docked vesicles has been unclear. This is in large part due to the observed variability in the positioning of the $CPX_{acc}$ (*Choi et al., 2016*; *Malsam et al., 2020*; *Zhou et al., 2017*;

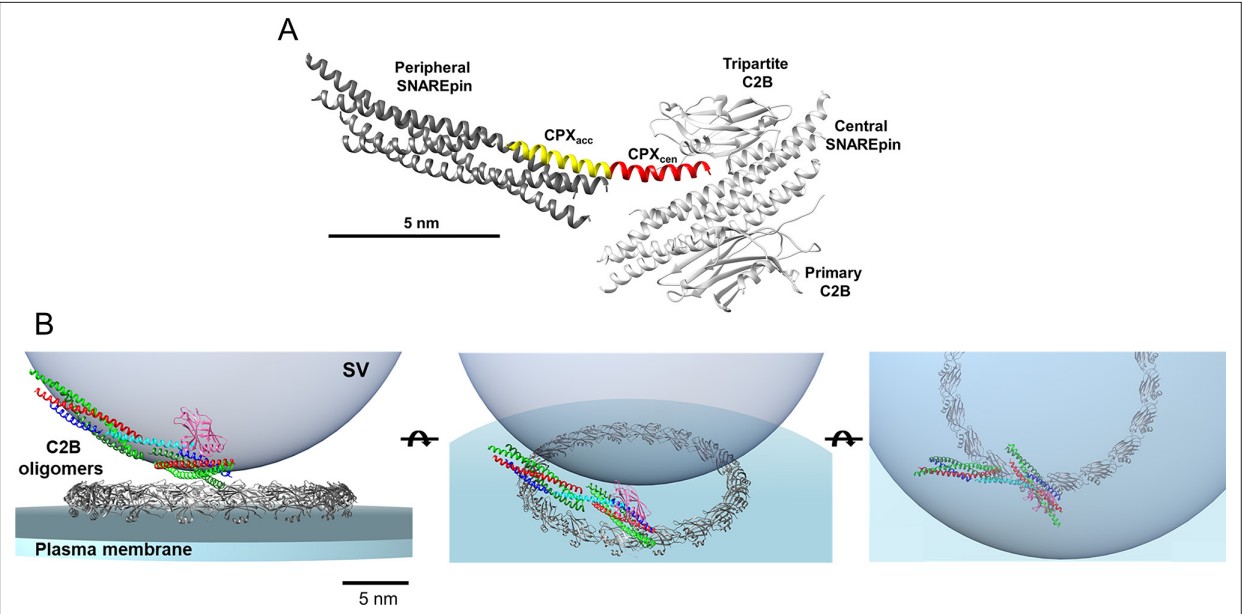

**Figure 5.** Synergistic regulation of SNARE-mediated fusion by CPX and Syt1. (**A**) Model of pre-fusion CPX-Syt-SNARE complex containing the *central* and *peripheral* SNAREpins connected via CPX trans-clamping interaction. The *central* SNAREpins, which are responsible for the $Ca^{2+}$-triggered fusion, are bound to and sterically clamped by two Syt molecules - one independently at the 'primary' interface and other in the conjunction with $CPX_{cen}$ (red) at the 'tripartite' interface. The $CPX_{acc}$ (yellow) emanating from the *central* SNAREs reaches out to bind and clamp the *peripheral* SNAREpin (dark gray). This molecular model was generated using the X-ray crystal structures 5W5C (*Zhou et al., 2017*) and 3RL0 (*Kümmel et al., 2011*) (see *Figure 5—figure supplement 1*). Noteworthy, the positioning of *peripheral* SNAREpins in this model is likely to be flexible considering the inherent variability in the localization of $CPX_{acc}$ (**B**) Organization of pre-fusion CPX-Syt-SNARE complex at the synaptic vesicle-plasma membrane interface. In addition to the 'bridging interaction', the primary C2B domain (gray) also self-assembles to an oligomeric structure which strengthens the Syt1 clamp on the central SNAREpins. The SNAREpins are multi-colored, CPX is cyan and tripartite C2B is pink. Only a single cross-linked SNAREpins is shown, but multiple SNARE complexes are likely involved in driving rapid SV fusion (see *Figure 5—figure supplement 2*). We have omitted the transmembrane domains of SNAREs/Syt and the Syt C2A domains for clarity.

The online version of this article includes the following figure supplement(s) for figure 5:

**Figure supplement 1.** X-ray structure 3RL0 (orange) representing the CPX trans-clamping interaction (Kummel et al.*Kümmel et al., 2011*) was superimposed onto the crystal structure 5W5C (gray) of the primed CPX-Syt1-SNARE complex (*Zhou et al., 2017*) to generate the 'bridging model' shown in *Figure 5A*.

**Figure supplement 2.** Possible organization of pre-fusion CPX-Syt-SNARE complexes under a docked vesicle.

*Kümmel et al., 2011*). $CPX_{acc}$ has been proposed to interact with c-terminal portion of the t-SNARE and VAMP2, both in a *cis* configuration that is $CPX_{cen}$ and $CPX_{acc}$ bound to the same SNAREpin (*Choi et al., 2016*; *Malsam et al., 2020*) or in a *trans* configuration that is $CPX_{cen}$ and $CPX_{acc}$ interacting with neighboring SNAREpins (*Choi et al., 2016*; *Kümmel et al., 2011*; *Krishnakumar et al., 2015*; *Cho et al., 2014*). We favor the *trans* insertion clamping model as this arrangement would enable CPX to regulate the distinct *central* and *peripheral* SNAREpin populations (*Figure 5*, see below).

Noteworthy, we observe that the specific interactions of the $CPX_{acc}$ with the synaptic SNARE proteins increase the potency of the clamp, and in accordance mCPX is ~2–3 fold more efficient in establishing the fusion clamp as compared to dmCPX or ceCPX under the same experimental conditions (*Figure 3*). However, the divergence in clamping ability among the mammalian, fly, and nematode CPXs is diminished at higher concentrations of CPX. This might explain the puzzling observation that in physiological analyses, when CPX is over-expressed, cross-species rescue experiments are largely successful yet $CPX_{acc}$-SNARE disrupting mutants' exhibit limited effect on the CPX clamping ability (*Yang et al., 2010*; *Cho et al., 2014*; *Radoff et al., 2014*). Considering that the $CPX_{acc}$ is highly divergent across different species, it is conceivable that $CPX_{acc}$ has distinctively evolved to optimally bind and clamp the species-specific SNARE partners. Additional biochemical/structural studies are needed to address this question.

Overall, our data strongly argues that mCPX has an intrinsic capacity to inhibit SNARE-dependent fusion and under minimal conditions is required (along with Syt1) to generate and maintain a pool of release-ready vesicles. Indeed, functionality of mCPX observed in our in vitro system perfectly matches with physiological studies in model invertebrate systems (*Martin et al., 2011*; *Hobson et al., 2011*; *Huntwork and Littleton, 2007*; *Cho et al., 2014*). However, recent physiological studies in mammalian synapses reported that acute CPX loss reduces SV fusion probability but does not unclamp spontaneous fusion. Hence, they conclude that CPX is dispensable for 'fusion clamping' in mammalian neurons (*López-Murcia et al., 2019*). It is worth noting that under these conditions CPX removal abates both spontaneous and evoked neurotransmitter release without changing the number of docked vesicles (*López-Murcia et al., 2019*). This suggests that acute CPX loss likely affects the late-stage vesicle priming process, and it is possible this 'loss-of-fusion' phenotype occludes CPX role in regulating spontaneous fusion events. Indeed, rescue experiments in CPX1/2/3 triple-knockout mouse background show that the CPX$_{acc}$ mutants enhances the spontaneous fusion events without altering evoked release, revealing that mCPX has a strong suppressive clamping function (*Malsam et al., 2020*). It is feasible mCPX also plays a more specialized role in mammalian synapses and is primarily involved in stabilizing newly primed synaptic vesicles and prevents their premature fusion (*Dhara et al., 2014*; *Chang et al., 2015*). In doing so, mCPX may function as a fusion clamp in an activity-dependent manner and is critical to blocking spontaneous/tonic and asynchronous vesicular release (*Dhara et al., 2014*; *Chang et al., 2015*; *Yang et al., 2010*) and indirectly promoting synchronous SV exocytosis.

mCPX on its own is ineffective in clamping SNARE-driven vesicle fusion, as the c-terminal portion of VAMP2 assembles into the SNARE complex far faster than free CPX can bind to prevent its zippering (*Gao et al., 2012*). As such, a delay in SNARE zippering is required for the CPX to bind and thereby block fusion. The fact that sufficient delay can be artificially provided by ~20 copies of DNA duplexes (*Figure 1*) suggest that under physiological conditions, Syt1 (and perhaps other proteins on the SV) might hinder the SNARE assembly by a simple steric mechanism, enabling mCPX to function as a fusion clamp. This is supported by the observation that the Syt1 clamp or the formation of the *central* SNAREpins are not strictly required for mCPX clamping function (*Ramakrishnan et al., 2020*).

Ca$^{2+}$-activation studies (*Figure 4*) show that mCPX also contributes to Ca$^{2+}$-triggered vesicle fusion from the clamped state. Reinforcing our earlier reports (*Ramakrishnan et al., 2019*; *Ramakrishnan et al., 2020*), we find that Syt1 and a small number of SNAREs are largely sufficient to get Ca$^{2+}$-evoked fusion with ~70% of vesicles fusing in response to 100 µM Ca$^{2+}$. Inclusion of mCPX increases the fusion probability with >90% Ca$^{2+}$-triggered fusion from the clamped state (*Figure 4*). We do not observe any change in the fusion kinetics ($\tau$~11ms) without or with mCPX (*Figure 4*), at least with our current time resolution of ~13 ms and persistent high Ca$^{2+}$ levels as opposed to Ca$^{2+}$-transients in the synapse.

Deletion/mutational analyses reveal that the α-helical CPX$_{cen}$ and CPX$_{acc}$ are the minimal domain required for the activation function (*Figure 4*). Specifically, the well-defined CPX$_{cen}$-SNARE interactions (*Chen et al., 2002*) was found to be critical for the stimulatory function and this effect is observed even low VAMP2 conditions that is with vesicles containing Syt1-clamped *central* SNAREpins only (*Figure 4*). This is in line with our previous finding that CPX$_{cen}$ interaction with the SNAREs, independent of the clamping functionality, is important for Ca$^{2+}$-evoked release in *Drosophila* neuromuscular junctions (*Cho et al., 2014*). Our data shows that CPX$_{acc}$ also contributes to the activation function, but the underlying mechanism is unclear. CPX$_{acc}$ could act indirectly by promoting CPX$_{cen}$ binding (*Radoff et al., 2014*) *or* directly by interacting with the SNARE complex albeit in a manner different from the clamping interactions.

The data presented here, taken together with our earlier report (*Ramakrishnan et al., 2020*), suggests a parsimonious model of how Syt1 and CPX could regulate SNARE-mediated fusion (*Figure 5*, *Figure 5—figure supplement 1*, *Figure 5—figure supplement 2*). We posit that under every docked vesicle, there are two types of SNAREpins – the *central* SNAREpins which are bound to Syt1 and are responsible for Ca$^{2+}$-triggered release and *peripheral* SNAREpins which are not bound to Syt1 and thus, not directly regulated by Ca$^{2+}$. We further suggest that the *central* and *peripheral* SNAREpins are equal in number and are assembled as a pair via a common, bridging molecule of CPX (*Figure 5A*). At the early stages of SV docking, Syt1 oligomers bind and clamp sub-set of *central* SNAREpins via the 'primary' interface (*Ramakrishnan et al., 2020*). CPX bind the Syt1-associated *central* SNAREpins via the CPX$_{cen}$ which positions the CPX$_{acc}$ helix to bind the t-SNAREs

an oppositely-oriented SNAREpin occupying the space where the C-terminal half of VAMP2 would ordinarily zipper to drive fusion. In this way, CPX$_{acc}$ acts to clamp the *peripheral* SNAREpin. This 'bridging model' (*Figure 5*, *Figure 5—figure supplement 1*) is based on the known 'trans-clamping' interaction observed in the pre-fusion CPX/SNAREpin crystal structure (*Kümmel et al., 2011*) and is validated by biochemical and functional analyses both previously (*Cho et al., 2014*; *Krishnakumar et al., 2015*; *Krishnakumar et al., 2011*; *Kümmel et al., 2011*) and in the current work. In addition, CPX$_{acc}$ might also directly interact with the *peripheral* VAMP2 c-terminus to prevent its assembly (not shown in *Figure 5*).

As evidenced in the recent crystal structure (*Zhou et al., 2017*), CPX$_{cen}$ binding to the *central* SNAREpins likely creates a new binding interface for second Syt1 to bind the same SNAREpins. Thus, mCPX could regulate Ca$^{2+}$ triggered vesicle fusion via the 'tripartite' interface (*Figure 5*, *Figure 5—figure supplement 1*, *Figure 5—figure supplement 2*). Supporting this proposition, we have previously shown the 'tripartite' interface is not necessary to produce stably docked vesicles but is required for efficient Ca$^{2+}$-triggered fusion from the clamped state (*Ramakrishnan et al., 2020*). In fact, disrupting binding of Syt1 to the tripartite interface lowers the fusion probability (~25%) similar to that observed with the removal of mCPX (*Ramakrishnan et al., 2020*). Furthermore, as the tripartite binding motif is largely conserved among different Synaptotagmin isoforms, so it is possible that mCPX binding could enable synergistically regulation of vesicular release by different calcium sensors (*Volynski and Krishnakumar, 2018*; *Zhou et al., 2017*). In addition to creating the 'tripartite' interface, mCPX binding might also promote vesicle fusion by stabilizing the full zippering SNARE complex. Obviously, this model is highly speculative and further functional studies (with higher temporal resolution, physiological Ca$^{2+}$ dynamics and different calcium sensors) as well as high-resolution structural data of vesicle-membrane junctions are needed to dissect the precise role of mCPX and its synergistic action with Syt1 in regulating Ca$^{2+}$-triggered vesicular fusion from the clamped state.

## Materials and methods
### Proteins and materials
The following cDNA constructs, which have been previously described (*Krishnakumar et al., 2013*; *Ramakrishnan et al., 2019*; *Ramakrishnan et al., 2020*), were used in this study: full-length VAMP2 (VAMP2-His$^6$, residues 1–116); full-length VAMP2$^{4X}$ (VAMP2-His$^6$, residues 1–116 with L70D, A74R, A81D, L84D mutations), full-length t-SNARE complex (mouse His$^6$-SNAP25B, residues 1–206 and rat Syntaxin1A, residues 1–288); Synaptotagmin (rat Synaptotagmin1-His$^6$, residues 57–421); Complexins (human His$^6$-Complexin 1, residues 1–134; *C. elegans* His$^6$-Complexin, residues 1–143; *Drosophila* His$^6$-Complexin1, residues 1–139). All mCPX mutants (truncations/point-mutations) were generated in the same background. All proteins were expressed and purified as described previously (*Krishnakumar et al., 2013*; *Ramakrishnan et al., 2019*; *Ramakrishnan et al., 2020*). All the lipids used in this study were purchased from Avanti Polar Lipids (Alabaster, AL). ATTO647N-DOPE was purchased from ATTO-TEC, GmbH (Siegen, Germany) and Calcium Green conjugated to a lipophilic 24-carbon alkyl chain (Calcium Green C24) was custom synthesized by Marker Gene Technologies (Eugene, OR). HPLC-purified DNA sequences (5'-ATCTCAATTATCCTATTAACC-3' and 5'-GGTTAATAGGATAATT GAGAT-3') conjugated to cholesterol with a 15 atom triethylene glycol spacer (DNA-TEG-Chol) were synthesized at Yale Keck DNA sequencing facility.

### Liposome preparation
VAMP2 ( ± Syt1) were reconstituted into small unilamellar vesicles (SUV) were using rapid detergent (1% Octylglucoside) dilution and dialysis method as described previously (*Ramakrishnan et al., 2019*; *Ramakrishnan et al., 2020*). The proteo-SUVs were further purified via float-up using discontinuous Nycodenz gradient. The lipid composition was 88 (mole) % DOPC, 10% PS and 2% ATTO647-PE for VAMP2 ( ± Syt1) SUVs and we used protein: lipid (input) ratio of 1:100 for VAMP2 for physiological density, 1: 500 for VAMP2 at low copy number, and 1: 250 for Syt1. Based on the densitometry analysis of Coomassie-stained SDS gels and assuming the standard reconstitution efficiency, we estimated the vesicles contain 73 ± 6 (normal physiological-density) or 13 ± 3 (low-density) and 25 ± 4 copies of outward-facing VAMP2 and Syt1 respectively (*Figure 1—figure supplement 1*).

## Single-vesicle fusion assay

All the single-vesicle fusion measurements were carried out with suspended lipid bilayers as previously described (*Ramakrishnan et al., 2018*; *Ramakrishnan et al., 2019*; *Ramakrishnan et al., 2020*). Briefly, t-SNARE-containing giant unilamellar vesicles (80% DOPC, 15% DOPS, 3% PIP2 and 2% NBD-PE) were prepared using the osmotic shock protocol and busted onto $Si/SiO_2$ chips containing 5 µm diameter holes in presence of HEPES buffer (25 mM HEPES, 140 mM KCl, 1 mM DTT) supplemented with 5 mM $MgCl_2$. The free-standing lipid bilayers were extensively washed with HEPES buffer containing 1 mM $MgCl_2$ and the fluidity of the t-SNARE containing bilayers was verified using fluorescence recovery after photo-bleaching using the NBD fluorescence.

Vesicles (100 nM lipids) were added from the top and allowed to interact with the bilayer for 10 min. The ATTO647N-PE fluorescence introduced in the vesicles were used to track vesicle docking, post-docking diffusion, docking-to-fusion delays and spontaneous fusion events. The time between docking and fusion corresponded to the fusion clamp and was quantified using a 'survival curve' whereby delays are pooled together, and their distribution is plotted in the form of a survival function (Kaplan-Meier plots). For the end-point analysis, the number of un-fused vesicles (survival percentage) was estimated ~10 s post-docking. After the initial 10 min, the excess vesicles were removed by buffer exchange (3 x buffer wash) and 1 mM $CaCl_2$ was added from the top to monitor the effect of $Ca^{2+}$ on the docked vesicles. The number of fused (and the remaining un-fused) vesicles was estimated (end-point analysis) ~ 1 min after $Ca^{2+}$-addition. CPX protein (at the indicated final concentration) were added to the experimental chamber and incubated for 5 min prior to the addition of the vesicles. Note: Pre-incubation with either the bilayer or the vesicle does not affect the clamping ability of mCPX and we chose to use pre-incubation with the bilayer (prior to adding SUVs) for the sake of convenience (*Figure 1—figure supplement 2*). All experiments were carried out at 37 °C using an inverted laser scanning confocal microscope (Leica-SP5) and the movies were acquired at a speed of 150ms per frame, unless noted otherwise. Fate of each vesicles were analyzed using our custom written MATLAB script described previously (*Ramakrishnan et al., 2018*). The files can be downloaded at: https://www.mathworks.com/matlabcentral/fileexchange/66521-fusion-analyzer-fas.

## Single-vesicle docking analysis

To get an accurate count of the docked vesicles, we used VAMP2 mutant protein (L70D, A74R, A81D, and L84D; VAMP2$^{4X}$) that eliminates fusion without impeding the docking process (*Krishnakumar et al., 2013*). For the docking analyses, 100 nM VAMP2$^{4X}$ containing SUVs (vSUV$^{4X}$) were introduced into the chamber and allowed to interact with the t-SNARE bilayer for 10 min. The bilayer was then thoroughly washed with the running buffer (3 x minimum) and the number of docked vesicles were counted, using Image J software.

## DNA-regulated single vesicle fusion assay

To prepare ssDNA containing vesicles, dialyzed VAMP2 or t-SNARE containing SUVs were incubated with complementary DNA-TEG-Chol for 2 hr at room temperature with mild-shaking. The v-SUVs were further purified using the Nycodenz gradient. We used the lipid: DNA-TEG-Chol input ratios of 1:2000, 1:1000, 1:500, and 1: 200 produce vSUVs with approximately 5, 10, 20, 50 copies of ssDNA per vesicles respectively. To identify the optimal condition for the single-vesicle fusion assays, we first tested the fusogenicity of ssDNA containing vesicles using bulk-fusion assay (*Figure 1—figure supplement 2*). Fusion of vSUV with t-SNARE liposomes were un-affected up to 20 copies of ssDNA, but we observed some reduction in fusion levels with 50 copies of ssDNA (*Figure 1—figure supplement 2*). Correspondingly, in the single-vesicle fusion setup, vSUV with 5, 10, and 20 copies of ssDNA docked and fused spontaneously with progressive docking-to-fusion delays, but the majority of 50 ssDNA-vSUV remained docked and un-fused (Data not shown). So, we chose to test the effect of Cpx on 20 ssDNA-vSUV, with 5 ssDNA-vSUV as the control.

## Calcium dynamics

We used a high-affinity $Ca^{2+}$-sensor dye, Calcium Green ($K_d$ of ~75 nM) conjugated to a lipophilic 24-carbon alkyl chain (Calcium Green C24) introduced in bilayer to monitor the arrival of $Ca^{2+}$ (100 µM). To estimate the arrival of $Ca^{2+}$ at or near the docked vesicle precisely, as indicated by increased in Calcium green fluorescence at 532 nm, we used resonant scanner to acquire movies at a speed of

up to 13 ms per frame with 512 × 32 resolution. For each vesicle fusion kinetics, calcium arrival was monitored over area of an individual hole (5 µm diameter) to get the high signal-to-noise ratio and vesicle fusion was monitored with 0.5 µm ROI around the docked vesicle. In these experiments, we used Sulforhodamine-B loaded Syt1-vSUV and tracked full-fusion events using increase in fluorescence signal due to dequenching of Sulforhodamine-B.

## Acknowledgements

This work was supported by National Institute of Health (NIH) grant DK027044. We thank Dr. Kirill Grushin for help with the structural models.

## Additional information

### Funding

| Funder | Grant reference number | Author |
| --- | --- | --- |
| National Institute of Diabetes and Digestive and Kidney Diseases | DK027044 | Shyam S Krishnakumar<br>James E Rothman |

The funders had no role in study design, data collection and interpretation, or the decision to submit the work for publication.

### Author contributions

Manindra Bera, Conceptualization, Formal analysis, Investigation, Methodology, Writing – original draft, Writing – review and editing; Sathish Ramakrishnan, Conceptualization, Formal analysis, Investigation; Jeff Coleman, Methodology, Resources; Shyam S Krishnakumar, Conceptualization, Formal analysis, Investigation, Supervision, Writing – original draft, Writing – review and editing; James E Rothman, Conceptualization, Investigation, Supervision, Writing – review and editing

### Author ORCIDs

Manindra Bera ⬤ http://orcid.org/0000-0001-9297-8126
Sathish Ramakrishnan ⬤ http://orcid.org/0000-0002-7844-2234
Shyam S Krishnakumar ⬤ http://orcid.org/0000-0001-6148-3251

### Decision letter and Author response

Decision letter https://doi.org/10.7554/eLife.71938.sa1
Author response https://doi.org/10.7554/eLife.71938.sa2

## Additional files

### Supplementary files

• Transparent reporting form
• Source data 1. Combined data and summary statistics for all figures and figure supplements.

### Data availability

All data generated or analyzed during this study are included in the manuscript and supporting files. Source data files for Figures 1, 2, 3, 4, and gel blots associated supplements are provided.

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
