## [Editor Report]

Bera and colleagues revisit several mechanistic questions mainly centered on the accessory helix of mouse complexin (mCpx) and its contribution to the 'fusion clamp' property of mCpx whereby mCpx-SNARE interactions prevent full assembly and subsequent membrane fusion. This clamping function is believed to help generate a metastable pool of release-ready vesicles at the synapse, and it has been studied in a wide variety of systems including mouse, fly, worm, squid, fish, and diverse in vitro biochemical preps over the past ~ 20 years. The authors derive several conclusions from their efforts, but most relevant is a reiteration of a previous proposal that the accessory helix region of mCpx stabilizes a pre-fusion clamped state via interactions with SNAREs.

---

## [Decision Letter]

**Decision letter after peer review:**

Thank you for submitting your Research Advance "Molecular Determinants of Complexin Clamping in Reconstituted Single-Vesicle Fusion" for consideration by *eLife*. Your article has been reviewed by 3 peer reviewers, and the evaluation has been overseen by a Reviewing Editor and Vivek Malhotra as the Senior Editor. The reviewers have opted to remain anonymous.

While we find the work potentially interesting, there are 3 concerns (see below) that should be addressed .

Essential revisions:

1. The authors report that Cpx can clamp fusion in the presence of 20 ssDNAs, while no clamping was observed with 5 ssDNAs. Since ssDNA is an synthetic factor, conclusions about the copy number of ssDNAs should be drawn with caution. Please perform titration experiments with various ratios of SNAREs and Cpx to demonstrate the clamping kinetics at a variety of protein ratios and concentrations.

2. The calcium concentration used in the experiments presented in Figure 4 is very high (1 millimolar). Thus, the "activating" effect of Cpx (established in several previous studies in neuronal cultures and in reconstitution experiments) could may have been masked by the high calcium concentration. Please repeat these experiments at a more physiological relevant calcium concentration (e.g., 10-100 μM).

3. For adding Cpx, the authors used the protocol listed as: "CPX protein (at the indicated final concentration) were added to the experimental chamber and incubated for 5 minutes prior to the addition of the vesicles." Thus, Cpx was incubated with plasma membrane mimicking bilayers. However, previous work has shown that Cpx also interacts with synaptic vesicles and that this interaction may be essential for its function. Moreover, since Cpx is a curvature sensor, as washed by the injection flow of the vesicles, the effective Cpx concentration used in the present experiments may be much lower than claimed. Moreover, as shown in their previous publication [Ramakrishnan 2020 *eLife*], Cpx can be washed off. Please co-incubate Cpx with vesicles, and then inject calcium to trigger fusion.

4. The authors utilize several past lines of evidence to support the fusion clamp view of Cpx function including older studies of RNAi knock-down of mouse Cpx1/2, studies from model invertebrates such as worm and fly, as well as the authors' own earlier in vitro work. In general, work on Cpx in cultured mammalian neurons has failed to observe large increases in spontaneous SV fusion rates over the past ~8 years and the field has moved away from relying on RNAi knock down of Cpx1/2 due to artefacts associated with Cpx3 expression. Thus, the authors are advised to be cautious about using the older siRNA KD data as an argument supporting an inhibitory role for Cpx of spontaneous release in neurons. Regardless, the inhibitory role of Cpx is well established in reconstituted systems where calcium-independent fusion is monitored (which is potentially distinct from spontaneous release in neurons that actually depends on low levels of calcium and may depend on other fusion sensors than synaptotagmin1/2). Please comment and discuss.

5. Conclusions from the lack of an observed Cpx effect on the speed of calcium-triggered fusion may be a bit overstated. There could be changes that are not being detected but are nonetheless physiologically relevant. Synapses display a variety of kinetics with distinct speeds ranging from 100 seconds of microseconds to milliseconds, so there is room for interesting kinetics well below the current 13 milliseconds cutoff of these experiments. Perhaps there are interesting Cpx effects in that physiological range. Please comment and discuss.

6. The temporal resolution for fast fusion experiments is 13 milliseconds. Thus, it is questionable to use a 7 millisecond time interval for the plots shown in Figure 4C. Moreover, in Figure 4A, the dotted lines to indicate the half-maximum of Sulforhodamine B jump between two time points. However, the signal of Sulforhodamine B reaches its maximum at the next time point. Therefore, the definition of "fusion time" is unclear. The single vesicle fusion assay used in this manuscript is based on the previous *eLife* article where the authors stated that "We typically observed the fluorescence signal increase at the bilayer surface between about three frames (~ 100 milliseconds) after calcium addition (Figure 2—figure supplement 1). We therefore used 100 milliseconds as the benchmark to accurately estimate the time-constants for the ca^2+^-triggered fusion reaction." Why the authors used the much shorter 7 millisecond time interval in this study? Please clarify and discuss.

7. For the purpose of delaying fusion long enough to allow the complexin clamp to form, the authors replace synaptotagmin with duplex DNA. Unlike the complexin/synaptotagmin machine, however, this system once clamped cannot proceed to fusion, so it is unclear if the duplex DNA clamped state resembles the physiologically clamped state. Since no physiological experiments are performed, the claims drawn by the present studied should be qualified.

*Reviewer #1 (Recommendations for the authors):*

1. The authors utilize several past lines of evidence to support the fusion clamp view of Cpx function including older studies of RNAi knock-down of mouse Cpx1/2, studies from model invertebrates such as worm and fly, as well as the authors' own earlier in vitro work. In general, work on Cpx in cultured mammalian neurons has failed to observe large increases in spontaneous SV fusion rates over the past ~8 years and the field has moved away from relying on RNAi knock down of Cpx1/2 due to artefacts associated with Cpx3 expression. I believe that continued use of the older siRNA KD data as an argument supporting an inhibitory role for Cpx doesn't seem like a constructive way forward when aiming for the mammalian Cpx research community to digest and assimilate this new in vitro data.

2. There were no mentions of fly or worm Cpx constructs in the Methods section. For instance, was fly Cpx a CAAX box variant? Could that explain its poor clamping? And there was no description of the Cpx(EAAK) construct. The current lack of relevant Materials documentation is not acceptable for publication.

3. Conclusions from the lack of an observed Cpx effect on the speed of calcium-triggered fusion may be a bit overstated. There could be changes that are not being detected but are nonetheless physiologically relevant. Synapses display a variety of kinetics with distinct speeds ranging from 100s of microseconds to milliseconds, so there's room for interesting kinetics well below the current 13 msec cutoff of these experiments. Perhaps there are interesting Cpx effects in that physiological range.

4. For the helix-breaking experiments performed in this manuscript using the GPGP insertion (Results section lines 194-200), there is a long history of past work on precisely this concept for Cpx clamping, but these were not directly mentioned or referenced. I would suggest including references to Chen 2002, Xue 2007, Cho 2014 and Radoff 2014 for their past work on destabilizing the propagation of secondary structure from the accessory to central helix and its requirement for Cpx inhibitory function.

*Reviewer #2 (Recommendations for the authors):*

1. The authors report that Cpx can clamp fusion in the presence of 20 ssDNAs, while no clamping was observed with 5 ssDNAs. Since ssDNA is an artificial factor, one cannot get any useful information from the copy number ssDNAs. Therefore, a titration experiment with various ratios of SNAREs and Cpx should be performed to demonstrate clamping kinetics.

2. The Cpx truncations and 4A mutant used in Figure 2 have been shown to reduce spontaneous release before Ca++. What hasn't been shown is the difference on clamping kinetics for these Cpx variants. Therefore, a titration experiment is necessary.

3. The Ca++ concentration in Figure 4 is too high to test the effect of Cpx. At that high Ca++ concentration, the activation effect of Cpx could be overridden by Ca++. Since effective Ca++-triggered fusion has been observed with 10-100uM Ca++ by many in vitro proteoliposome assays, they have to test 10-100uM Ca++ with much more physiological relevance.

4. For adding Cpx, the authors used the protocol listed as: "CPX protein (at the indicated final concentration) were added to the experimental chamber and incubated for 5 min prior to the addition of the vesicles." Basically, Cpx was incubated with plasma membrane mimicking bilayers. However, many papers have shown that Cpx should interact with synaptic vesicles for its function. Moreover, since Cpx is a curvature sensor, as washed by the injection flow of the vesicles, their effective Cpx concentration would be much lower than what has been claimed. As shown in their previous publication [Ramakrishnan 2020 *eLife*], Cpx can be washed off. Therefore, the authors should co-incubate Cpx with vesicles, and then inject Ca++ to trigger fusion.

5. The temporal resolution for fast fusion experiments is 13msec. The foundation of using a 7 msec time interval for their plots shown in Figure 4C is not clear. No explanation was provided. There is another unclear definition about fusion time. In the Figure 4A, the authors tried to use dotted lines to indicate the half-maximum of Sulforhodamine B jump between two time points. However, the signal of Sulforhodamine B reaches its maximum at the next time point. Therefore, it's not clear about their definition on "fusion time". The single vesicle fusion assay used in this manuscript is based on their previous publications [Ramakrishnan 2019 Langmuir; Ramakrishnan 2020 *eLife*]. In the method section of their 2020 *eLife* paper, the authors stated that "We typically observed the fluorescence signal increase at the bilayer surface between about three frames (~100 msec) after ca^2+^ addition (Figure 2—figure supplement 1). We therefore used 100 msec as the benchmark to accurately estimate the time-constants for the ca^2+^-triggered fusion reaction." It would be good to learn why the authors start to use 7 msec to estimate fusion in this paper.

*Reviewer # 3 (Recommendations for the authors):*

There are many typographical and/or grammatical errors that the authors should fix. In Figure 2, the color-coding scheme should be consistent. That is, mCPX(48-134) should not be blue in panels A and C and yellow in panel B. In the bar graphs in panel C, the survival percentage 0 should be on, not above, the x-axis.

[Editors' note: further revisions were suggested prior to acceptance, as described below.]

Thank you for resubmitting your work entitled "Molecular Determinants of Complexin Clamping and Activation Function" for further consideration by *eLife*. Your revised article has been evaluated by Vivek Malhotra (Senior Editor) and a Reviewing Editor.

The manuscript has been much improved but there are some remaining issues that need to be addressed before acceptance, as outlined below:

It is gratifying that the activating effect of Cpx becomes apparent at 100 μm ca^2+^. At that concentration, Cpx does not alter the fusion kinetics. However, it is possible that there is an effect on the fusion kinetics at even lower calcium concentrations. Please perform an additional experiment at a lower ca^2+^ concentration, e.g., 25 μm ca^2+^.

Lines 150-169: how were some of the numbers presented in this paragraph (~80%, ~20%, ~5-fold, etc.) derived from the figures cited?

Lines 239-250: the discussion of the mCPX(EAAK) construct is unclear. How does this explain the ability of this construct to rescue clamping (50%, similar to dmCPX and ceCPX) at higher concentrations?

Line 269: How many VAMP2 are on a vesicle? Is it ~70 (text), 60 (as suggested by the notation Syt1-VAMP60 in Figure 4B/C), or ~74 (Figure 4 legend)?

Lines 275-276: The phrase: "Deletion of the N-terminal (mCPX26-134) or the C-terminal domain (mCPX26-83)…" is not quite right. The second of these constructs lacks the N-terminal and the C-terminal domains.

Lines 279-281: "In fact, CPXacc deletion or CPXcen modification lowered fusion probability even below that observed under no mCPX condition (Figure 4D) suggesting that the CPXcen and CPXacc domains are crucial for mCPX's stimulatory function." How does the second part of the sentence follow from the first?

Lines 311-313: how CPXacc could be "capable of directly capturing the VAMP2 C-terminus and simultaneously occupying its binding pocket on the t-SNARE." This would seem to be a notable feat for any sequence, to say nothing of a sequence that can be swapped out for distantly related ones like (EAAK)7. Please clarify.

Lines 239-250: The results with mCPX(EAAK) are intriguing/puzzling. The authors seem to suggest that (EAAK)7 has 30% sequence identify with mCPXacc – is that really true? And if not, how should we think about the ability of this construct to rescue clamping (50%, similar to dmCPX and ceCPX) at higher concentrations?

---

## [Author Response]

Essential revisions:1. The authors report that Cpx can clamp fusion in the presence of 20 ssDNAs, while no clamping was observed with 5 ssDNAs. Since ssDNA is an synthetic factor, conclusions about the copy number of ssDNAs should be drawn with caution. Please perform titration experiments with various ratios of SNAREs and Cpx to demonstrate the clamping kinetics at a variety of protein ratios and concentrations.

We have examined the clamping function of mCPX under different reconstitution conditions in the DNA-regulated fusion experiments. Specifically, we assessed the effect of varying the mCPX concentration, VAMP2 density, ssDNA numbers and select mCPX mutants (both individually and in combination). This data, included as Figure 1 Supplement 4 in the revised manuscript, is consistent with our conclusions regarding the need for delay in fusion kinetics for mCPX to arrest SNARE-driven fusion and the underlying molecular mechanisms.

2. The calcium concentration used in the experiments presented in Figure 4 is very high (1 millimolar). Thus, the "activating" effect of Cpx (established in several previous studies in neuronal cultures and in reconstitution experiments) could may have been masked by the high calcium concentration. Please repeat these experiments at a more physiological relevant calcium concentration (e.g., 10-100 μM).

As recommended, we have repeated the ca^2+^-triggered fusion experiments with 100 µM ca^2+^ and find that mCPX increases the probability of fusion from the clamped state (~90% with mCPX vs. ~70% without mCPX) but does not alter the kinetics of release (t~11 ms under all conditions). We further carried out deletion/mutation analysis to identify that the CPX_cen_ + CPX_acc_ constitute the minimal domain for the stimulatory function. Interestingly, we find that neither the continuous, rigid helical domain nor the CPX_acc_-SNARE clamping interaction are required for the activation function. This data is included as Figure 4 in the revised manuscript and the Title/Abstract/Result/Discussion section have been revised accordingly.

3. For adding Cpx, the authors used the protocol listed as: "CPX protein (at the indicated final concentration) were added to the experimental chamber and incubated for 5 minutes prior to the addition of the vesicles." Thus, Cpx was incubated with plasma membrane mimicking bilayers. However, previous work has shown that Cpx also interacts with synaptic vesicles and that this interaction may be essential for its function. Moreover, since Cpx is a curvature sensor, as washed by the injection flow of the vesicles, the effective Cpx concentration used in the present experiments may be much lower than claimed. Moreover, as shown in their previous publication [Ramakrishnan 2020 eLife], Cpx can be washed off. Please co-incubate Cpx with vesicles, and then inject calcium to trigger fusion.

In the development of the in vitro assay, we systematically tested the effect of pre-incubation of CPX with the bilayer or SUVs and found no difference in the kinetics or clamping ability. We chose to use pre-incubation with the bilayer (prior to adding SUVs) for the sake of convenience. We have additionally tested and confirmed this proposition by using mCPX mutants lacking the c-terminal domain (mCPX^26-83^) or carrying L117W mutation that enhances its interaction with lipid bilayer (mCPX^L117W^). This data is included as Figure 1 Supplement 2 in the revised manuscript and explicated in the Methods section.

4. The authors utilize several past lines of evidence to support the fusion clamp view of Cpx function including older studies of RNAi knock-down of mouse Cpx1/2, studies from model invertebrates such as worm and fly, as well as the authors' own earlier in vitro work. In general, work on Cpx in cultured mammalian neurons has failed to observe large increases in spontaneous SV fusion rates over the past ~8 years and the field has moved away from relying on RNAi knock down of Cpx1/2 due to artefacts associated with Cpx3 expression. Thus, the authors are advised to be cautious about using the older siRNA KD data as an argument supporting an inhibitory role for Cpx of spontaneous release in neurons. Regardless, the inhibitory role of Cpx is well established in reconstituted systems where calcium-independent fusion is monitored (which is potentially distinct from spontaneous release in neurons that actually depends on low levels of calcium and may depend on other fusion sensors than synaptotagmin1/2). Please comment and discuss.

Our intent is to provide necessary background and highlight the ongoing debate on the clamping function of CPX, particularly in mammalian synapses. Thus, emphasizing the need for an in vitro reconstituted system to investigate the mechanistic details and complement the physiological studies. Indeed, we have clearly noted (in the same paragraph) that the observed differences in physiological studies could be due to perturbation method used and homeostatic compensatory mechanisms. We believe that we have adequately addressed this issue within the manuscript.

5. Conclusions from the lack of an observed Cpx effect on the speed of calcium-triggered fusion may be a bit overstated. There could be changes that are not being detected but are nonetheless physiologically relevant. Synapses display a variety of kinetics with distinct speeds ranging from 100 seconds of microseconds to milliseconds, so there is room for interesting kinetics well below the current 13 milliseconds cutoff of these experiments. Perhaps there are interesting Cpx effects in that physiological range. Please comment and discuss.

We have now repeated the experiment with physiologically-relevant ca^2+^ concentration (100 µM) and find that CPX increases the probability, but not the kinetics of fusion (See response to Q2 for details). It is possible that effect of CPX on kinetics of ca^2+^-activated fusion might be occluded by the time-resolution (13 ms) of our current experimental setup. We have mentioned this caveat in the Discussion section of the revised manuscript.

6. The temporal resolution for fast fusion experiments is 13 milliseconds. Thus, it is questionable to use a 7 millisecond time interval for the plots shown in Figure 4C. Moreover, in Figure 4A, the dotted lines to indicate the half-maximum of Sulforhodamine B jump between two time points. However, the signal of Sulforhodamine B reaches its maximum at the next time point. Therefore, the definition of "fusion time" is unclear.

We use the initial jump in Sulforhodamine-B and Calcium-green signal (marked by dashed lines) rather the peak fluorescence to define the fusion time as this likely correlate to earliest instance of fusion pore opening. Since this occurs within a single frame of recording, we used t_1/2_ (~7 msec) of our single frame (13 msec) for the plots in Figure 4C. We strongly believe that the fusion occurs <10 msec in these conditions.

7. The single vesicle fusion assay used in this manuscript is based on the previous eLife article where the authors stated that "We typically observed the fluorescence signal increase at the bilayer surface between about three frames (~ 100 milliseconds) after calcium addition (Figure 2—figure supplement 1). We therefore used 100 milliseconds as the benchmark to accurately estimate the time-constants for the ca^2+^-triggered fusion reaction." Why the authors used the much shorter 7 millisecond time interval in this study? Please clarify and discuss.

Consistent with our previous report, we observe the ca^2+^ fluorescence signal at the bilayer surface ~100 msec after addition. The ca^2+^-addition to the top of the chamber is denoted by arrow in Figure 4A in the revised manuscript (Note: This arrow was inadvertently placed at the 100 msec mark in our older version of the Figure 4A. We apologize for the confusion)

8. For the purpose of delaying fusion long enough to allow the complexin clamp to form, the authors replace synaptotagmin with duplex DNA. Unlike the complexin/synaptotagmin machine, however, this system once clamped cannot proceed to fusion, so it is unclear if the duplex DNA clamped state resembles the physiologically clamped state. Since no physiological experiments are performed, the claims drawn by the present studied should be qualified.

We used DNA-regulated fusion assay to assess and confirm that the delay in fusion is necessary for mCPX to arrest SNARE assembly. We have now carried out systematic titration of VAMP2, ssDNA and CPX concentration (including CPX mutants) to establish that the clamping mechanisms observed under DNA-regulated fusion reaction is identical to those observed in the presence of Syt1 (see response to Q1 for details). This allows us to directly infer that under physiologically-relevant conditions delay in fusion kinetics, likely imparted by Syt1 or other SV proteins, is required for CPX to clamp SNARE-mediated fusion.

[Editors' note: further revisions were suggested prior to acceptance, as described below.]

The manuscript has been much improved but there are some remaining issues that need to be addressed before acceptance, as outlined below:It is gratifying that the activating effect of Cpx becomes apparent at 100 μm ca^2+^. At that concentration, Cpx does not alter the fusion kinetics. However, it is possible that there is an effect on the fusion kinetics at even lower calcium concentrations. Please perform an additional experiment at a lower ca^2+^ concentration, e.g., 25 μm ca^2+^.

We find that the vast majority of the ca^2+^-triggered vesicle fusion occurs within a single frame irrespective of the ca^2+^-concentration (0.1 mM or 1 mM) or acquisition rate (13 msec or 36 msec). Furthermore, we observe that the mutations in Syt1 or CPX alter the probability of fusion, but not the fusion kinetics. This suggests that the fusion rate is likely much faster than the temporal resolution (13 msec) of our experimental setup. As such, any alterations in fusion kinetics, even at lower (25 mM) ca^2+^ concentration, might not be resolved in the current setup. We are working to increase the temporal resolution (1-2 ms) of our fusion system and to include better ca^2+^ control. Our future work is focused on using the upgraded fusion setup to dissect the role of Syt1/CPX in modulating vesicle fusion kinetics and as such, is beyond the scope of the current work.

Lines 150-169: how were some of the numbers presented in this paragraph (~80%, ~20%, ~5-fold, etc.) derived from the figures cited?

The fusion data for Syt1-vSUVs (black curve) is derived from the survival analysis shown in Figure 2B. Consistent with our previous reports (Ramakrishnan et al. 2019 and 2020), we observe that ~80% of Syt1-vSUVs fuse within 4-5 sec and remainder (~20%) stay ‘clamped’ and remain ca^2+^-sensitive up to 2-3 hours.

The fold changes in the number of docked vesicles are derived from the docking analysis shown in Figure 2A. We inadvertently included the wrong data for the no CPX condition in Figure 2A. We have corrected this mistake and have revised the fold-increase values in the manuscript. For example, we observe ~2-3 docked vesicles per 5 mm hole for Syt1-vSUVs (grey bar) and the inclusion of mCPX^WT^ (red bar) increases the number of docked vesicles to ~7-8 vesicles per 5 mm hole, i.e. ~3-fold increase in the number of docked vesicles.

Lines 239-250: the discussion of the mCPX(EAAK) construct is unclear. How does this explain the ability of this construct to rescue clamping (50%, similar to dmCPX and ceCPX) at higher concentrations?

Based on the sequence alignment of the accessory helical region alone (mCPX residues 26-48) we stated that mCPX^EAAK^ has 30% sequence similarity to mCPX_._ We acknowledge that this might be over-simplification as the accessory helix alignment is likely shaped by the central helix also. Indeed, if the central helix region is included (mCPX residues 26-70) in the sequence alignment, then the mCPX^EAAK^ accessory helix region has low-degree sequence similarity with the mCPX_acc_ (Figure 3 Supplement 3). Nonetheless, we find that the mCPX^EAAK^ accessory helix is more hydrophobic in nature and in fact, resemble couple of gain-of-function ‘super-clamp’ mutations (residues Asp-27 and Glu-34 replaced by Ala) that has been shown to increase the overall clamping efficiency (Figure 3 Supplement 3). This could possibly explain mCPX^EAAK^ ability to partially rescue clamping at 20 mM, similar to dmCPX and ceCPX, even though it was an artificial accessory helix sequence.

We have included the sequence alignment data as Figure 3 Supplement 3 in the revised manuscript. We have also revised the relevant section (Page 8) to remove the reference to 30% sequence identity and included the following language “We note that the accessory helix of mCPX^EAAK^ is more hydrophobic in nature and indeed, resembles couple of the gain-of-function ‘super-clamp’ mutations, with residue Asp-27 and Glu-34 replaced with Ala (Figure 3 Supplement 3). This could potentially explain how a randomized accessory helix sequence could partially rescue clamping at high (20 mM CPX) concentration.”

Line 269: How many VAMP2 are on a vesicle? Is it ~70 (text), 60 (as suggested by the notation Syt1-VAMP60 in Figure 4B/C), or ~74 (Figure 4 legend)?

We sought to reconstitute 60 copies of VAMP2 per vesicles, but based on the densitometry analysis, we estimate that each vesicle contains 73 ± 6 of outward facing VAMP2. To be consistent, we now denote as ~70 VAMP2 (text) *or* as VAMP^70^ (Figure 4 and Figure 1 Supplement).

Lines 275-276: The phrase: "Deletion of the N-terminal (mCPX26-134) or the C-terminal domain (mCPX26-83)…" is not quite right. The second of these constructs lacks the N-terminal and the C-terminal domains.

We apologize for this error and have fixed the language in the revised manuscript.

Lines 279-281: "In fact, CPXacc deletion or CPXcen modification lowered fusion probability even below that observed under no mCPX condition (Figure 4D) suggesting that the CPXcen and CPXacc domains are crucial for mCPX's stimulatory function." How does the second part of the sentence follow from the first?

We agree. This sentence is a confusing and we have revised this section (Page 9) to read as follows: “Deletion of the CPX_acc_ in addition to N-terminal domain (mCPX^48-134^) or disrupting the CPX_cen_-SNARE interaction (mCPX^4A^) abrogated the mCPX activation function (Figure 4D) suggesting that the CPX_cen_ and CPX_acc_ domains are crucial for mCPX’s stimulatory function”.

Lines 311-313: how CPXacc could be "capable of directly capturing the VAMP2 C-terminus and simultaneously occupying its binding pocket on the t-SNARE." This would seem to be a notable feat for any sequence, to say nothing of a sequence that can be swapped out for distantly related ones like (EAAK)7. Please clarify.

Our data, taken together with the previous studies (Kummel et al. 2011, Krishnakumar et al. 2011, Malsam et al. 2020) strongly indicate that the CPX_acc_ binds to c-terminal end of both VAMP2, and t-SNAREs and these interactions are part of the clamping mechanism. This dual-binding modality might be feasible considering that the c-terminal portion of the SNARE proteins, esp. VAMP2 is likely unstructured and flexible in the pre-fusion clamped state.